# SARS-CoV-2 Detection in Fecal Sample from a Patient with Typical Findings of COVID-19 Pneumonia on CT but Negative to Multiple SARS-CoV-2 RT-PCR Tests on Oropharyngeal and Nasopharyngeal Swab Samples

**DOI:** 10.3390/medicina57030290

**Published:** 2021-03-20

**Authors:** Barbara Brogna, Carlo Brogna, Mauro Petrillo, Adriana Modestina Conte, Giulio Benincasa, Luigi Montano, Marina Piscopo

**Affiliations:** 1Department of Radiology, Moscati Hospital, Contrada Amoretta, 83100 Avellino, Italy; 2Specialist-Craniomed Laboratory Group srl, Viale degli Astronauti, 45, 83038 Montemiletto, Italy; dir.brogna@craniomed.it; 3European Commission, Joint Research Centre (JRC), Ispra, 21027 Via Enrico Fermi, Italy; mauro.petrillo@ec.europa.eu; 4Chief U.O. Emergency Unit-OBI P.O., Pineta Grande Hospital, Via Domitiana 3000, 81030 Castel Volturno, Italy; adriana.conte@pinetagrande.it; 5Chief Anatomy Pathology Department, Pineta Grande Hospital, Via Domitiana 3000, 81030 Castel Volturno, Italy; giulio.benincasa@pinetagrande.it; 6Specialist Andrology Unit, Service of Lifestyle Medicine in Uro-Andrology, Local Health Authority (ASL), 84121 Salerno, Italy; l.montano@aslsalerno.it; 7Department of Biology, University of Naples Federico II, 80138 Naples, Italy; marina.piscopo@unina.it

**Keywords:** COVID-19, SARS-CoV-2, RT-PCR, feces, oropharyngeal swab, nasopharyngeal swab, fecal swab, viral pneumonia, chest, computed tomography

## Abstract

Reverse transcriptase polymerase chain reaction (RT-PCR) negative results in the upper respiratory tract represent a major concern for the clinical management of coronavirus disease 2019 (COVID-19) patients. Herein, we report the case of a 43-years-old man with a strong clinical suspicion of COVID-19, who resulted in being negative to multiple severe acute respiratory syndrome coronavirus 2 (SARS-CoV-2) RT-PCR tests performed on different oropharyngeal and nasopharyngeal swabs, despite serology having confirmed the presence of SARS-CoV-2 IgM. The patient underwent a chest computed tomography (CT) that showed typical imaging findings of COVID-19 pneumonia. The presence of viral SARS-CoV-2 was confirmed only by performing a SARS-CoV-2 RT-PCR test on stool. Performing of SARS-CoV-2 RT-PCR test on fecal samples can be a rapid and useful approach to confirm COVID-19 diagnosis in cases where there is an apparent discrepancy between COVID-19 clinical symptoms coupled with chest CT and SARS-CoV-2 RT-PCR tests’ results on samples from the upper respiratory tract.

## 1. Introduction

Coronavirus disease 2019 (COVID-19), caused by the severe acute respiratory syndrome coronavirus 2 (SARS-CoV-2), is a pandemic disease that can manifest with fever, pneumonia and, in severe cases, with acute respiratory distress symptoms (ARDS) [1,2]. Gastrointestinal symptoms with vomiting and diarrhea are often reported as other manifestations of the disease [3,4,5,6]. It is widely accepted that SARS-CoV-2 is mostly transmitted by respiratory droplets and fomite, and there is also evidence of fecal-oral transmission [1,3,4,5,6,7]. Asymptomatic and pauci-symptomatic individuals represent a major concern for the virus spread. Real-time reverse transcriptase polymerase chain reaction (RT-PCR) on oropharyngeal and nasopharyngeal (OP/NP) swab samples is considered the gold standard routine method for detection of SARS-CoV-2. However, SARS-CoV-2 RT-PCR tests giving negative results represent a diagnostic challenge for clinicians in the management of patients, especially when these negative results do not confirm clinical manifestations. In this context, computed tomography (CT) represents the most effective tool to diagnose pneumonia, and it can aid in supporting the diagnosis of COVID-19 in symptomatic cases, in the presence of multiple negative RT-PCR results, when conducted together with the SARS-CoV-2 serological tests [8,9,10]. However, recent studies reported that the virus can persist for a long time in feces, and it has been proposed to perform SARS-CoV-2 RT-PCR testing on fecal specimen as part of routine analyses for the detection of SARS-CoV-2, especially before the release of COVID-19 hospitalized patients [7,11,12,13,14,15,16,17,18,19].

## 2. Case Presentation

A 43-years-old patient with a contact history with COVID-19 patients came to the emergency room of our hospital because of the worsening of dyspnea, chills, and fever (38.5 °C). Upon clinical examination, he showed a normal pressure value PA 120/75 mmHg, cardiac palpitations, and diffuse reduced vesicular breathing. Arterial blood gas revealed respiratory failure with O2 saturation (SaO2) not higher than 90%. Blood tests revealed: mild leukopenia (3000/mm^3^), thrombocytopenia (96,000/mm^3^,), mild increase of the D- Dimer (0.7 mg/L), mild high levels of transaminases; all other blood tests were within their normal ranges and are listed in Table 1.

The patient reported that fever had started 12 days before and in the first days was associated to dyspnea, diarrhea, and myalgia. On the second day of fever, a RT-PCR testing on OP/NP wab sample (Allplex™ SARS-CoV-2/FluA/FluB/RSV Assay, Seegene, Italian distributor Arrow Diagnostics S.r.l, Genova (ITA)) was conducted by the local health authority and the result was negative. Due to the initial presence of diarrhea and the emerging studies that revealed the persistence of SARS-CoV-2 in the stool, the family doctor prescribed a SARS-CoV-2 RT-PCR test on the stool sample, which was carried out in an authorized testing laboratory. The RT- PCR on stool (adapted Allplex™ SARS-CoV-2/FluA/FluB/RSV Assay) resulted to be positive. The family doctor administered metronidazole (200 mg, 3 times/die) for one day. At the hospital, two days after an additional negative SARS-CoV-2 RT-PCR testing on an OP/NP swab (Easy^®^ SARS-CoV-2), it was agreed to perform a chest CT scan, due also to the worsening of the dyspnea, coupled with a SARS-CoV-2 serological test. The chest CT scan showed typical COVID-19 findings with areas of ground glass in a crazy paving pattern with consolidation in the right inferior lobe (Figure 1a,b) in a peripheral posterior distribution and with small areas of consolidation in the left inferior lobe (Figure 1c) and a small consolidation in the right superior lobe (Figure 1d).

The serological test, done using a chemiluminescent assay (Abbott Alinity i SARS-CoV-2 IgG) detected the presence of SARS-CoV-2 IgM (34.07 AU/mL) and the absence of SARS-CoV-2 IgG (cut-off 1.4 Index S/C as suggested by manufacturer).

As the patient’s symptoms worsened, it was decided to perform an additional SARS-CoV-2 RT-PCR testing on an OP/NP swab sample (Easy^®^ SARS-CoV-2), but the result was again negative. Due to the previous positive RT-PCR results for SARS-CoV-2 in the stool sample, the observation of pneumonia by chest CT scan, and the concomitant serological SARS-CoV-2 IgM positivity, at the hospital it was decided to initiate a therapeutical treatment with azithromycin (Azithromycin 500 mg/day, intravenous administration), corticosteroid (Methylprednisolone 400 mg/dL, intravenous administration), and low molecular weight heparin (4000 UI), in addition to oxygen therapy. After three days, the clinical conditions of the patient improved. At the end of this period, a SARS-CoV-2 RT-PCR test on OP/NP and faecal swab samples (Allplex™ SARS-CoV-2/FluA/FluB/RSV Assay) were conducted, which resulted in being negative and positive, respectively. A chest CT was repeated a week later which showed a reduction of the consolidations previously reported (Figure 2). Therefore, a week later another fecal swab sample (Allplex™ SARS-CoV-2/FluA/FluB/RSV Assay) was also repeated that finally resulted negative.

## 3. Discussion

Several researchers pointed out the importance of chest CT in presence of negative SARS-CoV-2 RT-PCR tests results, especially in the pandemic context, where the risk of underestimating the disease can increase the risk of virus transmission among individuals [9,20,21,22]. The Fleishner Society guidelines [8] suggests that chest imaging can be helpful in the patients triage in the presence of high pre-test probability. High sensitivity (67–100%) and relatively low specificity (25–80%) are reported for the CT scans [23], and CT specificity is higher only in the presence of high prevalence of the disease [23]. In the context of COVID-19, the Fleishner Society suggests the use of chest imaging in the presence of moderate and severe COVID-19 symptoms, in the follow-up of patients at high risk of pneumonia progression, in worsening patients, and in symptomatic patients when SARS-CoV-2 RT-PCR test results are negative, or not available, and in the presence of a high probability pre-test [8]. COVID-19-associated pneumonia has specific radiological image patterns consisting of ground glass areas (GGO), consolidation, and crazy paving motifs with a multifocal and posterior distribution or central and peripheral distribution. Vascular enlargement and bronchial distortion are also other reported radiological characteristics [24]. These radiological patterns vary according to the phase of the disease [21,25,26]. However, cases with normal CT scans in patients who resulted positive for SARS-CoV-2 RT-PCR in OP/NP swabs [27] have also been reported. For this reason, the current guidelines [8,28,29] suggest that chest CT cannot be used as a screening tool to diagnose COVID-19. Although RT-PCR on OP/NP swabs is considered the standard method to diagnose SARS-CoV-2, negative false results ranging from 1% to 30% [30,31]. The bronchoalveolar lavage fluid (BALF) specimen test might be the most accurate method, but the risk of exposure for healthcare personnel is relatively high [32]. It has been reported that in more than 30% of SARS-CoV-2 RT-PCR tests, confirmed COVID-19 patients have detectable SARS-CoV-2 RNA in stool, and recent studies revealed that viral RNA can be found in fecal specimens [19,33,34]. In addition, there is indication that SARS-CoV-2 RNA can persist more in stool than in the upper respiratory tract, and that patients with both severe and mild disease showed viral SARS-CoV-2 RNA in the fecal sample for more than four weeks after symptom onset [12,13,14,15,16,17,18,19]. Tang et al. [17] have also described the case of a child diagnosed as a COVID-19 patient thanks to a positive SARS-CoV-2 RT-PCR test on his fecal sample, in contrast to a negative SARS-CoV-2 RT-PCR tests result on the child’s OP/NP samples. Performing a SARS-CoV-2 RT-PCR test on stool of symptomatic patients with repetitive negative SARS-CoV-2 RT-PCR tests results can be used as an additional rapid test to support diagnosis by chest CT coupled with a serological test, or to support diagnosis by serological tests only, when chest CT is not possible. Our case report is in line with the collection of findings made by Neu et al. [35], who reviewed evidence of interactions between viruses and gut bacteria, and how these interactions can have a fundamental role in the pathogenic phase against the eukaryotic cells. It is also coherent with what was described by G. Petruk et al. [36] about the identification of interaction between SARS-CoV-2 spike protein and E. Coli lipopolysaccharides and with the identification of the persistence of SARS-CoV-2 nucleic acids and immunoreactivity in the intestinal biopsies (small bowels) obtained from asymptomatic individuals four months after the onset of COVID-19 [37].

## 4. Conclusions

The here reported case provides additional evidence that the use of SARS-CoV-2 RT-PCR tests on stool will help to confirm diagnosis of COVID-19 in highly suspicious cases, especially when SARS-CoV-2 RT-PCR tests on OP/NP swabs end with negative results. These tests can be used to support, or in combination with, CTs and serological tests to speed-up COVID-19 diagnosis.

## Figures and Tables

**Figure 1 medicina-57-00290-f001:**
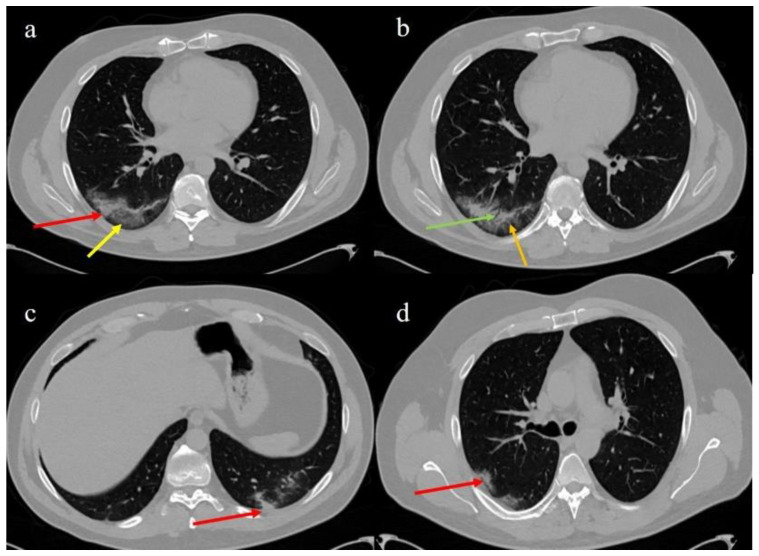
Chest computed tomography (CT) images of the patient with the typical distribution of COVID-19 pneumonia: (**a**) consolidation (red arrow) and crazy paving area (yellow arrow) with a posterior and peripheral distribution in the right inferior lobe; (**b**) observed air bronchogram (green arrow) in the consolidation together with small vascular vessel enlargement (orange arrow); (**c**) small consolidations in the left inferior lobe (red arrow); (**d**) small consolidations in the posterior segment of the right superior lobe (red arrow).

**Figure 2 medicina-57-00290-f002:**
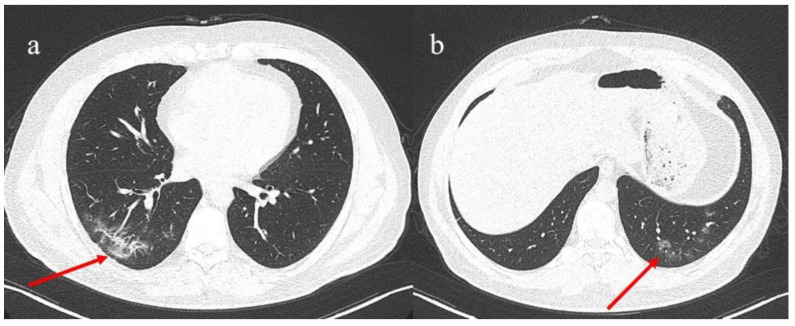
On the left panel (**a**), reduction of the consolidation the right inferior lobe (red arrow) after therapy; on the right panel (**b**), the reduction of areas previously observed as small zones of consolidations with some ground glass (GGO) area (red arrow) in the absorptive phase after therapy.

**Table 1 medicina-57-00290-t001:** Main laboratory analyses result with the reporting systemic unit (SU) of measurements at the hospital admission and the normal value range.

Laboratory Parameters	SU	Patient’s Value at Hospital Admission	Normal Value Range
Hemoglobin	mg/dL	16.0	13.0–17.5
Mean cell volume	fL	89.1	80.0–98.0
Platelets count	X1000/μL	96.0	140.00–450.00
White blood count	X1000/μL	3.00	4.00–11.0
Neutrophils	%	57.6	40.0–75.0
Lymphocytes	%	33.7	20.0–50.0
Monocytes	%	6.1	0.0–11.0
Eosinophils	%	0.4	0.0–0.7
Basophiles		0.6	0.0–0.2
Aspartate transaminases	U/L	40	<37
Alanine transaminases	U/L	44	<41
Glycemia	mg/dL	90	60–110
Creatinine	mg/dL	1.01	0.7–1.3
Lactate dehydrogenase	U/L	196	135–225
C Reactive Protein	mg/dL	0.47	<0.5
D-Dimer	mg/dL	0.7	<0.3

## Data Availability

Data available on request due to restrictions eg privacy or ethical.

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
