# Peer review of "SARS-CoV-2 Detection in Fecal Sample from a Patient with Typical Findings of COVID-19 Pneumonia on CT but Negative to Multiple SARS-CoV-2 RT-PCR Tests on Oropharyngeal and Nasopharyngeal Swab Samples"

_medicina, 2021, doi:10.3390/medicina57030290_

Round 1

Reviewer 1 Report

Highly well written case report. Please let me give some minor comments.

Case presentation:

  1. Is an expression "with a positive epidemiological history" a widely accepted term in Italy? It seems better to rewrite as "with a contact history of COVID-19 patients."
  2. If it is allowed to add table, it will be better to summarize all blood tests data in to one table with normal value.
  3. "Due to the initial presence of diarrhea, the family doctor performed the same SARS-CoV-2 RT-PCR test on stool sample thatresulted to be positive." Is it a normal practice if negative results were detected from OP/NP samples? It seems better to add a sentence (or phrase) how the physician suspected Covid-19 from diarrhea (e.g. based on case reports or guidelines or etc).
  4. If you add (a)(b)(c)(d), you need to present your explanations. Or you can delete this classification, and remain the caption as it is.
  5. Quality of Figure 3 is very low, and if the paper is accepted in the future, it may dame the reputation of the paper itself. Considering the strength of this paper, I recommend the authors to increase the resolution of figure 3.

Discussion:

  1. "negative false results are reported in up to 50% of cases (30)."
    It seems you have cited the report (?) in the early phase of the pandemic. Considering your case report has been focusing on the symptomatic cases, you need to cite several systematic articles that measured the sensitivity of RT-PCR testing.
  2. "It has been reported that 32%-67% of SARS-
    CoV-2 RT-PCR tests confirmed COVID-19 patients have detectable SARS-CoV-2 RNA in
    stool, and recent studies revealed that viral RNA can be found in faecal specimens (12-19)."
    How did you conclude "32%-67%"? It seems you cited several papers of case reports and etc but this expression is not accurate. You need to cite the result of several systematic reviews.

Conclusion:

  1. The expression "is a straightforward solution" is rather exaggerated. You need to re-write as "will help" or etc.

Author Response

REVIEWER 1

Dear Reviewer,

thank you for your valuable comments and suggestions that will increase the quality of the manuscript. Please find below details of what we have changed in the manuscript to accomplish your comments.

CASE PRESENTATION

  1. Is an expression "with a positive epidemiological history" a widely accepted term in Italy? It seems better to rewrite as "with a contact history of COVID-19 patients."

Thank you for your observations. It has been amended in the text: “A 43-years-old patient with a contact history of COVID-19 patients […]”

  1. If it is allowed to add table, it will be better to summarize all blood tests data in to one table with normal value.

Thank you for this suggestion. We have created the Table 1 with the patient’s laboratory value at the Hospital admission and with the range of the normal value

  1. Due to the initial presence of diarrhea, the family doctor performed the same SARS-CoV-2 RT-PCR test on stool sample that resulted to be positive." Is it a normal practice if negative results were detected from OP/NP samples? It seems better to add a sentence (or phrase) how the physician suspected Covid-19 from diarrhea (e.g. based on case reports or guidelines or etc).

Thank you for this suggestion. It has been amended in the text: “Due to the initial presence of diarrhea and the emerging studies that revealed the persistence of SARS-CoV-2 into stool,[…]”

  1. If you add (a)(b)(c)(d), you need to present your explanations. Or you can delete this classification, and remain the caption as it is.

Thank you for your suggestions. It has been amended in the text and in the figure

  1. Quality of Figure 3 is very low, and if the paper is accepted in the future, it may dame the reputation of the paper itself. Considering the strength of this paper, I recommend the authors to increase the resolution of figure 3.

Thank you for this observation. We have deleted it and we specify in the text when the RT-PCR on stool resulted negative in the text.

DISCUSSION

  1. "negative false results are reported in up to 50% of cases (30)."
    It seems you have cited the report (?) in the early phase of the pandemic. Considering your case report has been focusing on the symptomatic cases, you need to cite several systematic articles that measured the sensitivity of RT-PCR testing.

Thank you for your suggestions we have cited other 2 articles and edited the sentence.

`           References:

  1. Long, DR; Gombar, S; Hogan, CA; Greninger, AL; O’Reilly-Shah, V; BrysonCahn, C; et al. Occurrence and timing of subsequent SARS-CoV-2 RT-PCR positivity among initially negative patients. Clin. Infect. Dis. 2020. doi: 10.1093/cid/ciaa722
  2. Del Campo, R.; Zamora, J et al. False-negative results of initial RT-PCR assays for COVID-19: a systematic review. PloS one 2020, 15, e0242958.

  1. It has been reported that 32%-67% of SARS-CoV-2 RT-PCR tests confirmed COVID-19 patients have detectable SARS-CoV-2 RNA in stool, and recent studies revealed that viral RNA can be found in faecal specimens (12-19)." How did you conclude "32%-67%"? It seems you cited several papers of case reports and etc but this expression is not accurate. You need to cite the result of several systematic reviews.

We agree with you. We edited the text and added 2 other citations (systematic reviews).

References

  1. Rokkas, T. Gastrointestinal involvement in COVID-19: a systematic review and meta-analysis. Ann. Gaestrenterol. 2020, 33, 355. doi: 10.20524/aog.2020.0506
  2. Cheung, K. S.; Hung, I. F.; Chan, P. P.; Lung, K. C.; Tso, E.; Liu, Leung, W. K et al. Gastrointestinal manifestations of SARS-CoV-2 infection and virus load in fecal samples from a Hong Kong cohort: systematic review and meta-analysis. Gastroenterology 2020, 159, 81-95. doi: 10.1053/j.gastro.2020.03.065.)

CONCLUSIONS

  1. The expression "is a straightforward solution" is rather exaggerated. You need to re-write as "will help" or etc

Thank you for your suggestion it has been amended in the text

Reviewer 2 Report

The illustrated case report is very interesting, even if the report is not a novelty in scientific literature.

This reviewer believes that some minor changes are required for a better understanding of the manuscript.

Case presentation section.
Please specify the enzymatic activity of AST and ALT and their reference intervals.

Can the authors explain how it's possible that the family doctor did a molecular investigation for SARS-CoV-2?

Italian doctors have all the devices (biohazard hoods and nucleic acid extraction and RT-PCR systems) in their offices?

Or did the doctor prescribe only the examination and the analysis was carried out in an authorized laboratory? 

It is essential that the authors indicate the RT-PCR analysis system and reagents used  for SARS-CoV-2 detection on the stool sample; is this device CE-IVD certified for use in diagnostics or is an RUO (research use only) product?

Regarding the SARS-CoV-2 IgM test, please specify the name and manufacturer of the test and its cut-off.

Author Response

Dear Reviewer,

Many thanks for your valuable support and suggestions that will increase and enhance the

manuscript’s quality. We have applied them and you can find all the modifications marked in red

Case presentation section.

  1. Please specify the enzymatic activity of AST and ALT and their reference intervals

Thank you for your observations. We have created a table (Table 1) and reported also the value of AST and ALT

  1. Can the authors explain how it is possible that the family doctor did a molecular investigation

for SARS-CoV-2?

Thank you for this observation and we are sorry for the misunderstanding. The first RT-PCR was made by the Health Local Authority that in Italy are named ASL. This type of Health organization is present only in Italy and we have previously reported to family doctor to simplify the case description. Anyhow, to better clarify it, It has been edited in the text, i.e. “[…] was made by the health local authority and the result was negative”

  1. Can the authors explain how it is possible that the family doctor did a molecular investigation for SARS-CoV-2? Italian doctors have all the devices (biohazard hoods and nucleic acid extraction and RT-PCR systems) in their offices? Or did the doctor prescribe only the examination and the analysis was carried out in an authorized laboratory? 

Thank you for your observations. The family doctor prescribed a RT-PCR on stool that was analyzed in an authorized laboratory. We reported these corrections in the text: The family doctor prescribed a SARS-CoV-2 RT-PCR test on stool sample that was carried out in an authorized laboratory.

  1. It is essential that the authors indicate the RT-PCR analysis system and reagents used for SARS-

CoV-2 detection on the stool sample; is this device CE-IVD certified for use in diagnostics or is

an RUO (research use only) product?

Thank you for this suggestion. We have added the name of the used kit in every of the Nasal-/Oro- pharyngeal, fecal swabs assays and for serology described performed assay. You can find it in the text marked in red. To our knowledge they are all CE-IVD devices and listed in the https://covid-19-diagnostics.jrc.ec.europa.eu/, currently considered reference in EU for tests and devices considered as appropriate for use in the context of COVID-19, and in line with countries’ testing strategies

(https://eur-lex.europa.eu/legal-content/EN/TXT/PDF/?uri=CELEX:52021DC0035&from=EN)

  1. Regarding the SARS-CoV-2 IgM test, please specify the name and manufacturer of the test and

its cut-off.

It has been amended in the text: The serological test, done using a chemiluminescent assay (Abbott

Alinity i SARS-CoV-2 IgG) detected the presence of SARS-CoV-2 IgM (34.07 AU/mL) and the absence

of SARS-CoV-2 IgG.
